# Impact of the cost-of-living crisis on the nature of attempts to stop smoking and to reduce alcohol consumption in Great Britain: A representative population survey, 2021–2022

Sarah E. Jackson[1,2]*, Sharon Cox[1,2], Lion Shahab[1,2], Jamie Brown[1,2]

1 Department of Behavioural Science and Health, University College London, London, United Kingdom,
2 SPECTRUM Consortium, London, United Kingdom

* s.e.jackson@ucl.ac.uk

**Data Availability Statement:** All data used for these analyses are available on Open Science Framework (https://osf.io/yv3p7/).

## Abstract

### Background

Smoking and excessive drinking place a strain on household budgets. We aimed to examine the impact of the cost-of-living crisis in Great Britain on the nature of smoking cessation and alcohol reduction attempts, and explore changes in health professionals offering support.

### Methods

Data were from 14,567 past-year smokers and high-risk drinkers (AUDIT-C ≥5) participating in monthly representative surveys, January-2021 through December-2022. We estimated time trends in cost as a motive driving the most recent (smoking cessation/alcohol reduction) attempt, use of paid or evidence-based support, and receipt of GP offer of support for smoking cessation or alcohol reduction, and tested for moderation by occupational social grade.

### Results

The proportion of attempts motivated by cost did not change significantly over time among smokers (25.4% [95%CI = 23.8–26.9%]), but increased between December-2021 and December-2022 among high-risk drinkers from less advantaged social grades (from 15.3% [95%CI 12.1–19.3] to 29.7% [20.1–44.1]). The only change in support use was an increase in smokers using paid support, specifically e-cigarettes (from 28.1% [23.7–33.3] to 38.2% [33.0–44.4]). Among those visiting their GP, the proportion who received an offer of support was similar over time among smokers (27.0% [25.7–28.2]) and high-risk drinkers (1.4% [1.1–1.6%]).

### Conclusions

There is limited evidence that the 2021/22 cost-of-living crisis affected the nature of attempts to stop smoking and reduce alcohol consumption, or receipt of GP offer of support. It is encouraging that use of evidence-based support has not declined and that use of e-

**Funding:** Cancer Research UK (PRCRPG-Nov21 \100002) funded data collection and SJ and SC's salaries. The funders had no role in study design, data collection and analysis, decision to publish, or preparation of the manuscript.

**Competing interests:** JB has received unrestricted research funding from Pfizer and J&J, who manufacture smoking cessation medications. LS has received honoraria for talks, an unrestricted research grant and travel expenses to attend meetings and workshops from Pfizer, and has acted as paid reviewer for grant awarding bodies and as a paid consultant for health care companies. All authors declare no financial links with tobacco companies, e-cigarette manufacturers, the alcohol industry, or their representatives. This does not alter our adherence to PLOS ONE policies on sharing data and materials.

cigarettes in quit attempts has increased. However, cost is increasingly motivating alcohol reduction attempts among less advantaged drinkers, and rates of GPs offering support, especially for alcohol reduction, remain very low.

## Introduction

There is a cost-of-living crisis in Great Britain, and smoking and alcohol use remain urgent issues. Tobacco smoking and excessive alcohol consumption are not only leading causes of disease and premature death [1, 2], they also place a significant burden on household budgets [3]. Although concerns for health are usually the primary motive for changing these behaviours, cost is also an important factor for many smokers and high-risk drinkers [4, 5]. A range of evidence-based aids are available to increase chances of successfully quitting smoking or reducing alcohol consumption; some are available free of charge for those with recourse to some public benefits (e.g., medications on prescription, behavioural support) and others are available to purchase (e.g., over-the-counter [OTC] nicotine replacement therapy [NRT], e-cigarettes). Understanding how the recent cost-of-living crisis in Great Britain is affecting the nature of attempts to stop smoking and reduce alcohol consumption, including the importance of cost as a motive and use of support, is important for informing the provision of support to those who want (or need) to stop smoking or consume less alcohol in the current financial climate.

The cost-of-living crisis refers to a fall in 'real' disposable incomes (that is, adjusted for inflation and after taxes and benefits) that people living in Great Britain have experienced since late 2021 [6]–although many working class families have struggled with household budgets for much longer than this [7]. High rates of inflation have seen the cost of everyday essentials like groceries and household bills rising faster than average household incomes. Causes include a range of global and local factors, including the Covid-19 pandemic, Russia's invasion of Ukraine, Brexit, and national tax increases [6, 8]. The cost-of-living crisis may have affected the nature of attempts to stop smoking and to reduce alcohol consumption in two key ways. First, it may have caused a shift in the motives driving quit attempts. As people's disposable income has decreased, smokers and drinkers may have increasingly considered quitting or cutting down to save money. Recent data from the Smoking Toolkit Study show the average smoker in England spends £20 a week on cigarettes [9], equivalent to 3.3% of gross weekly earnings (or almost two weeks' income each year) for a person earning the median salary in the UK [10] and 5.3% (or three weeks' income each year) for a person working a 40-hour week on minimum wage [11]. Quitting smoking has been described as an 'instant pay rise' that could help ease the financial struggles many smokers are currently facing [12]. Data from the 2020/21 Living Costs and Food Survey show the average UK household spends £11 a week on alcohol [13] (which likely underestimates drinkers' expenditure, given ~20% of UK adults do not drink at all [14]), indicating similar potential to make savings by reducing alcohol consumption. However, it is also possible that people have switched to cheaper products to reduce their expenditure without changing their consumption. Secondly, the cost-of-living crisis may have affected the ways in which smokers try to quit and drinkers try to cut down. The two most popular aids used by smokers trying to quit are e-cigarettes (used in ~30% of quit attempts) and over-the-counter NRT (used in ~10%) [15]; both of which cost the user money. It is possible that smokers have been less likely to use cessation support for which they have to pay as their disposable incomes have decreased. If this is happening, it is important to know because unsupported quit attempts are less likely to be successful [16], and there are a wide

range of free cessation support options smokers could be directed to as an alternative. It is possible that health professionals are already responding to the cost-of-living crisis by increasing their efforts to offer smokers who want to quit and high-risk drinkers who want to reduce their consumption support to do so.

In examining its effects on attempts to stop smoking and reduce alcohol consumption, it is important to consider that the cost-of-living crisis is unlikely to have affected all smokers and drinkers equally. In particular, any impact is likely to have been greater among socioeconomically disadvantaged groups. There is a social gradient in smoking, whereby low-income earners and those who have access to state support and live in social housing are more likely to smoke and have higher levels of dependence than those who are more affluent [17, 18]. While overall alcohol consumption is typically less among lower earners [14], half of UK households living in poverty buy alcohol each week, spending a median of £13 a week (5.6% of weekly disposable income) and experience higher mortality risk [19]. With lower levels of disposable income, socioeconomically disadvantaged groups are also being hit harder by the rising costs of living in Great Britain (e.g., energy, fuel, food): almost half (42%) of people living in the most deprived quintile of areas in England reported cutting back on food and essentials between March and June 2022 because their cost of living had increased, compared with 27% in the least deprived quintile [20]. While socioeconomically disadvantaged smokers are just as motivated to quit as those from more advantaged groups, they tend to be less successful in stopping even when they use evidence-based support [18], so identifying any differences in the impact of the cost-of-living crisis may be important for targeting support.

Another factor that may moderate the impact of the cost-of-living crisis on attempts to stop smoking is the use of manufactured vs. hand-rolled cigarettes. Smokers who mainly smoke manufactured cigarettes spend around twice as much on average as those who mainly smoke hand-rolled cigarettes (£28 vs. £16 per week [9]), which may make cost a more urgent motive for quitting or paying for cessation support less affordable. It is therefore plausible that any impact of the cost-of-living crisis on quit attempts may be more pronounced among smokers who mainly smoke manufactured cigarettes. However, it may also be the case that people already smoking hand-rolled cigarettes are more vulnerable to cost pressures, and therefore may be more likely to try to quit smoking as the cost of living increases.

This study therefore aimed to examine the impact of the cost-of-living crisis in Great Britain on the nature of smokers' attempts to quit smoking and high-risk drinkers' attempts to reduce alcohol consumption (specifically, cost as a motive and use of paid and evidence-based support), and explore differences by key moderators (socioeconomic position [indexed by occupational social grade] and main type of cigarettes smoked). A secondary aim was to explore whether there has been any impact of the cost-of-living crisis on health professionals offering support for smoking cessation and alcohol reduction.

## Method

### Pre-registration

The analysis plan was pre-registered on Open Science Framework (https://osf.io/yv3p7/). We made one amendment, which included extending the study period to include data through December 2022 to provide up-to-date estimates (analyses on data through July 2022, as originally specific in the analysis plan, are available at https://osf.io/yv3p7/). Following peer review, we added an unplanned analysis of trends in attempts to stop smoking and reduce alcohol consumption over the study period, for comparison with trends in attempts that were motivated by cost.

## Design

Data were drawn from the ongoing Smoking and Alcohol Toolkit Studies, monthly cross-sectional surveys of a representative sample of adults in Great Britain [21–23]. The studies use a hybrid of random probability and simple quota sampling to select a new sample of approximately 2,400 adults (≥18y) each month. Data are collected via computer-assisted telephone interviews. Full details of the sampling procedure are provided elsewhere [23]. All participants are asked questions on both smoking and alcohol use. Since April 2022, the majority of items on alcohol use (with the exception of Alcohol Use Disorders Identification Test—consumption (AUDIT-C)) have only been included in alternate waves for participants in England due to availability of funding (they have continued to be collected in all waves for participants in Scotland and Wales).

Comparisons of data collected face-to-face (before the Covid-19 pandemic necessitated a switch to telephone interviews) with other national surveys indicate that key variables such as key sociodemographic variables, smoking prevalence, and cigarette consumption are nationally representative [22]. The telephone-based data collection uses the same combination of random location and quota sampling, and weighting approach as the face-to-face interviews and comparisons of the two data collection modalities indicate good comparability [24].

## Population

The present study used data collected between January 2021 (around a year before the cost-of-living crisis began to affect Great Britain [6]) and December 2022 (the most recent data available at the time of analysis).

We used data from participants who reported (i) being a current cigarette smoker or having stopped smoking in the past year ('past-year smokers') or (ii) high-risk drinking, defined by an AUDIT-C score ≥5 ('high-risk drinkers'). We were only able to analyse drinking outcomes among high-risk drinkers (as opposed to all adult drinkers) because our outcomes of interest were not assessed in participants reporting low or moderate consumption (i.e., AUDIT-C score <5). Analyses of the nature of attempts to quit smoking or reduce alcohol consumption and moderation of changes in these outcomes focused on those who reported having made at least one serious attempt to quit smoking (past-year smokers) or reduce their alcohol consumption (high-risk drinkers) in the past year. Analyses of receipt of GP offer of support and moderation of changes in these outcomes focused on those who reported having seen their GP in the last year.

## Ethics approval and consent to participate

Ethical approval was provided by the UCL Research Ethics Committee (0498/001). Participants provide informed consent to take part in the study, and all methods are carried out in accordance with relevant regulations. The data are not collected by UCL and are anonymised when received by UCL.

## Measures

**Outcomes.** Past-year smokers and high-risk drinkers who reported having made a serious attempt to quit smoking/reduce their alcohol consumption in the last year were asked a series of questions about their most recent attempt.

*Cost as a motive* for trying to quit smoking/reduce alcohol consumption was assessed [among past-year smokers/high-risk drinkers, respectively] with the question: 'Which of the following, if any, do you think contributed to you making the most recent [quit attempt/

attempt to restrict your alcohol consumption]?' Participants could select multiple motives from a list of options. Those who responded 'A decision that [smoking/drinking] was too expensive' were coded 1, else they were coded 0.

Use of support for smoking cessation and alcohol reduction was assessed with the question: 'Which, if any, of the following did you try to help you [stop smoking/restrict your alcohol consumption] during the most recent attempt?' Participants could select multiple cessation aids from a list of options.

*Use of paid support for smoking cessation* was coded 1 for past-year smokers who reported using an e-cigarette or NRT product (e.g., patches/gum/inhaler) without a prescription, else it was coded 0. Use of these aids was also analysed separately to see whether any overall changes in use of paid cessation aids was driven by one aid in particular.

*Use of any evidence-based support for smoking cessation* was coded 1 for past-year smokers who reported using an e-cigarette, NRT on prescription, varenicline, or bupropion; attending stop smoking group or one or more stop smoking one-to-one counselling/advice/support session(s) (either in person or remotely); or using telephone support, a website, an app, or written self-help materials, else it was coded 0. OTC NRT was not included given a lack of clear evidence that it increases the success rate of quit attempts [16, 25].

*Use of any evidence-based support for alcohol reduction* was coded 1 for high-risk drinkers who reported using prescription medication (e.g., acamprosate, disulfiram, or nalmefene); attending a specialist alcohol clinic or centre or one or more one-to-one or group counselling/advice/support sessions for help with drinking (either in person or remotely); consulting a community pharmacist for help with drinking; or using telephone support, a website, an app, or written self-help materials, else it was coded 0.

*Receipt of GP offer of support for smoking cessation* was assessed with the question: 'Has your GP spoken to you about smoking in the past year (i.e. last 12 months)?

1. Yes, he/she suggested that I go to a specialist stop smoking adviser or group;

2. Yes, he/she suggested that I see a nurse in the practice;

3. Yes, he/she offered me a prescription for Champix, Zyban, a nicotine patch, nicotine gum or another nicotine product;

4. Yes, he/she suggested that I use an e-cigarette;

5. Yes, he/she advised me to stop but did not offer anything;

6. Yes, he/she asked me about my smoking but did not advise me to stop smoking;

7. No, I have seen my GP in the last year but he/she has not spoken to me about smoking;

8. No, I have not seen my GP in the last year.'

Those who responded 'yes' were able to select multiple responses between 1 and 6 to indicate all types of advice they received. Those who responded 'no' were able to select only one response option (7 or 8). Receipt of offer of support was coded 1 for those who selected response options 1–4 and 0 for those who selected response options 5–7. Those who responded that they had not seen their GP in the last year (response 8) were excluded from analyses of this outcome.

*Receipt of GP offer of support for alcohol reduction* was assessed with the question: 'In the last 12 months, has a doctor or other health worker within your GP surgery discussed your drinking?

1. Yes, a doctor or other health worker within my GP surgery asked about my drinking;

2. Yes, a doctor or other health worker within my GP surgery offered advice about cutting down on my drinking;

3. Yes, a doctor or other health worker within my GP surgery offered help or support within the surgery to help me cut down;

4. Yes, a doctor or other health worker within my GP surgery referred me to an alcohol service or advised me to seek specialist help;

5. No, I have seen a doctor or health worker within my GP surgery in the last 12 months but did not discuss my drinking;

6. No, I have not seen a doctor or health worker within my GP surgery in last 12 months.'

Those who responded 'yes' were able to select multiple responses between 1 and 4 to indicate all types of advice they received. Those who responded 'no' were able to select only one response option (5 or 6). Receipt of offer of support was coded 1 for those who selected response options 3 or 4 and 0 for those who selected response options 1, 2, or 5. Those who responded that they had not seen a doctor or health worker within their GP surgery in the last year (response 6) were excluded from analyses of this outcome.

**Exposure.** *Survey month* was coded 1 (January 2021) through 24 (December 2022).

**Potential moderators.** *Occupational social grade* was categorised as ABC1, which included managerial, professional, and upper supervisory occupations and C2DE, which included manual routine, semi-routine, lower supervisory, and long-term unemployed. This is a valid classification that is widely used in research in UK populations [26].

*Main type of cigarettes smoked* was assessed with questions that asked past-year smokers to report the number of cigarettes they usually smoke(d) each day or each week (as preferred) and how many of these they think are hand-rolled. We defined hand-rolled cigarette smokers as those reporting at least 50% of their total cigarette consumption is hand-rolled and manufactured cigarette smokers as those reporting at least 50% is manufactured. This definition has been used in previous studies [27] and has the benefit of allowing inclusion of non-daily smokers and those who smoke both hand-rolled and manufactured cigarettes.

**Covariates.** In addition to the potential moderators, we included age, gender, and geographic region as covariates to account for any changes in the profile of past-year smokers or high-risk drinkers over the study period. *Age* was categorised as 18–24, 25–34, 35–44, 45–54, 55–65, or ≥65 years. *Gender* was self-reported as man, woman, or in another way. Those who identified in another way were excluded from analyses that adjusted for gender due to low numbers. *Region* was categorised as England, Scotland, or Wales.

## Statistical analysis

Data were analysed in R v.4.2.1. Data were weighted to match the demographic profile of Great Britain on age, social grade, region, housing tenure, ethnicity, and working status within sex [22]. Missing cases were excluded on a per-analysis basis.

For each outcome, we estimated the proportion and 95%CI and used log-binomial regression to test the association with survey month, adjusting for age, gender, region, and social grade. Analyses of cost as a motive for trying to quit smoking and use of support for smoking cessation were also adjusted for the main type of cigarettes smoked. Survey month was modelled using restricted cubic splines with three knots placed at the earliest, middle, and latest months, to allow relationships with time to be flexible and non-linear, while avoiding categorisation.

To explore moderation by social grade (analyses of all outcomes) and main type of cigarettes smoked (analyses of cost as a motive for trying to stop smoking and use of support for

smoking cessation), we repeated the relevant models including the interaction between the moderator of interest and survey month–thus allowing for time trends to differ across sub-groups (i.e., social grades C2DE vs. ABC1, and smokers who mainly use manufactured vs. hand-rolled cigarettes). Each of the interactions was tested in a separate model.

We used predicted estimates from our models to (i) plot the prevalence of each outcome over the study period (overall and by moderating variables) and (ii) obtain specific predictions for the prevalence of each outcome in December 2022 (the last month in the study period) compared with January and December 2021 (the first and middle months in the study period), incorporating information from all survey months and thus increasing statistical precision.

## Results

There were 54,360 respondents to the Smoking and Alcohol Toolkit Studies between January 2021 and December 2022, of whom 7,412 were current cigarette smokers, 1,206 had stopped smoking in the last year, and 17,475 were high-risk drinkers. High-risk drinkers surveyed in England in May, July, September, and November 2022 (n = 2,076) were ineligible for inclusion in our analysis as they were not asked questions on our outcomes of interest (alcohol questions were only assessed in alternate waves over this period due to availability of funding).

In the 12 months prior to the survey, 3,064 past-year smokers (41.3%) and 3,617 eligible high-risk drinkers (23.5%) reported making at least one serious attempt to stop smoking and reduce alcohol consumption, respectively, and 4,817 (65.0%) past-year smokers and 8,672 (56.3%) eligible high-risk drinkers reported visiting their GP. Table 1 summarises the weighted characteristics of these groups. Our analysed sample comprised 14,567 unique cases. There were complete data on all variables except main type of cigarettes smoked (missing in 228 [7.4%] smokers who tried to quit) and gender (missing in 16 [0.1%] participants).

### Cost as a motive for smoking cessation and alcohol reduction

Across the study period, 24.6% of past-year smokers who had tried to quit in the last year and 14.5% of high-risk drinkers who had made an alcohol reduction attempt in the last year cited cost as a motive driving their most recent attempt (Table 2).

Our models showed no significant change for smoking cessation attempts (Fig 1, Table 2), nor any significant differences in trends by social grade ($p_{interaction}$ = 0.208; Fig 2A) or main type of cigarettes smoked ($p_{interaction}$ = 0.480; S1 Fig). However, the raw weighted data suggested a potential increasing trend between the first and third quarters of 2022, before falling in quarter four (Fig 1).

There was an uncertain rise for alcohol reduction attempts in the latter half of the study period (p = 0.076), from 12.0% at the start of the cost-of-living crisis in December 2021 to 16.3% a year later (Fig 1, Table 2). The raw weighted data showed a similar pattern to smoking cessation attempts, with some indication of levelling off in the fourth quarter of 2022 (Fig 1)–at least among those from social grades ABC1 (Fig 2). Time trends differed significantly by social grade ($p_{interaction}$ = 0.044), with a significant rise between December 2021 and December 2022 among those from social grades C2DE (from 15.3% [95%CI 12.1–19.3] to 29.7% [20.1–44.1]) but little change among those from social grades ABC1 (Fig 2B). This rise was specific to alcohol reduction attempts that were motivated by cost, with no such increase observed when we analysed trends in all alcohol reduction attempts over this period (S2–S4 Figs).

### Use of support for smoking cessation and alcohol reduction

Across the study period, 45.4% of past-year smokers who had tried to quit in the last year reported using paid support in their most recent quit attempt (17.3% OTC NRT and 32.8% e-

**Table 1. Sample characteristics.**

| | Smokers who tried to quit[1] | Smokers who visited their GP[2] | High-risk drinkers who tried to reduce their alcohol consumption[3] | High-risk drinkers who visited their GP[4] |
|---|---|---|---|---|
| *Unweighted N* | *3064* | *4817* | *3617* | *8672* |
| Age (years) | | | | |
| 18–24 | 21.6% | 17.5% | 10.7% | 13.7% |
| 25–34 | 29.7% | 26.8% | 19.5% | 18.2% |
| 35–44 | 17.4% | 17.2% | 20.0% | 15.9% |
| 45–54 | 13.9% | 15.2% | 22.4% | 18.7% |
| 55–64 | 10.0% | 11.2% | 15.9% | 16.2% |
| 65+ | 7.5% | 12.0% | 11.4% | 17.2% |
| Gender | | | | |
| Men | 51.3% | 46.1% | 57.3% | 56.6% |
| Women | 47.6% | 52.8% | 42.0% | 42.6% |
| In another way | 1.1% | 1.1% | 0.7% | 0.8% |
| Social grade C2DE | 57.6% | 58.7% | 33.0% | 38.7% |
| Region | | | | |
| England | 87.8% | 87.1% | 85.3% | 81.9% |
| Wales | 3.6% | 4.7% | 4.9% | 5.9% |
| Scotland | 8.6% | 8.2% | 9.8% | 12.3% |
| Mainly smokes hand-rolled cigarettes | 47.9% | 51.9% | - | - |

Data are shown as weighted percentages.

There were some missing data for some variables (7.4% main type of cigarettes smoked among smokers who tried to quit; 0.1% gender). Valid percentages are shown.

[1] Current cigarette smokers or those who quit in the past year who reported having made at least one serious attempt to stop smoking in the last 12 months.

[2] Current cigarette smokers or those who quit in the past year who reported having visiting their GP in the last 12 months.

[3] High-risk drinkers (AUDIT-C $\geq$5) who reported having made at least one serious attempt to reduce their alcohol consumption in the last 12 months.

[4] High-risk drinkers (AUDIT-C $\geq$5) who reported having visiting their GP in the last 12 months.

cigarettes), and 41.0% reported using any evidence-based support (Table 2). 9.4% of high-risk drinkers who had tried to reduce their alcohol consumption in the last year reported using evidence-based support in their most recent attempt (Table 2; use of paid support in alcohol reduction attempts was not captured in the survey).

Between January 2021 and December 2022, there was a significant rise in the use of any paid support for smoking cessation (from 41.4% to 52.0%; p = 0.006; Fig 3, Table 2). When we analysed subtypes of paid support (OTC NRT and e-cigarettes) separately, we found this was driven by a rise in e-cigarette use (from 28.1% to 38.2%; p = 0.006), with no significant change in OTC NRT use (Fig 3, Table 2). There were no significant changes in use of evidence-based support in attempts to stop smoking or reduce alcohol consumption (Fig 4, Table 2).

Time trends in OTC NRT use differed significantly by social grade ($p_{interaction}$ = 0.006). The difference occurred between January and December 2021 (i.e., prior to the cost-of-living crisis), with a rise in OTC NRT use in quit attempts among smokers from social grades ABC1 (from 11.4% [8.2–15.9] to 19.2% [16.1–22.9]) but an uncertain decline among those from social grades C2DE (from 21.2% [15.3–29.3] to 16.6% [13.5–20.4]; S5 Fig).

Time trends in use of e-cigarettes ($p_{interaction}$ = 0.034) and evidence-based support ($p_{interaction}$ = 0.029) differed significantly by main type of cigarettes smoked. These differences occurred between December 2021 and December 2022 (i.e., during the cost-of-living crisis), with both e-cigarette use and use of any evidence-based support rising among those who mainly smoked hand-rolled cigarettes (from 31.2% [27.1–35.9] to 44.9% [36.9–54.7] and from

**Table 2. Raw outcome data aggregated across the study period (January 2021 –December 2022) and modelled estimates for the first, middle, and last months in the time series.**

| | Raw data[1] | | | Modelled estimates | | | | | | | | |
| --- | --- | --- | --- | --- | --- | --- | --- | --- | --- | --- | --- | --- |
| | (January 2021 – December 2022) | | | January 2021[2] (start of study period) | | | December 2021[2] (middle of study period) | | | December 2022[2] (end of study period) | | |
| | % | Lower CI | Upper CI | % | Lower CI | Upper CI | % | Lower CI | Upper CI | % | Lower CI | Upper CI |
| Smokers who tried to quit | | | | | | | | | | | | |
| Motivated by cost | 25.4 | 23.8 | 26.9 | 26.2 | 21.8 | 31.5 | 23.9 | 21.4 | 26.9 | 23.0 | 18.9 | 28.0 |
| Used paid support | 45.6 | 43.8 | 47.3 | 41.4 | 36.4 | 47.1 | 44.5 | 41.3 | 47.9 | 52.0 | 46.5 | 58.3 |
| Used OTC NRT | 18.4 | 17.0 | 19.8 | 16.5 | 12.9 | 21.1 | 17.6 | 15.3 | 20.3 | 16.3 | 12.8 | 20.7 |
| Used e-cigarette | 31.8 | 30.1 | 33.4 | 28.1 | 23.7 | 33.3 | 31.2 | 28.3 | 34.4 | 38.2 | 33.0 | 44.4 |
| Used evidence-based support | 40.9 | 39.2 | 42.6 | 39.8 | 34.8 | 45.5 | 40.1 | 37.0 | 43.5 | 43.2 | 37.9 | 49.2 |
| High-risk drinkers who tried to reduce their alcohol consumption | | | | | | | | | | | | |
| Motivated by cost | 13.9 | 12.8 | 15.0 | 12.5 | 9.9 | 15.8 | 12.0 | 10.4 | 13.9 | 16.3 | 12.2 | 21.9 |
| Used evidence-based support | 9.3 | 8.4 | 10.3 | 8.1 | 6.1 | 10.7 | 9.3 | 7.8 | 11.1 | 9.3 | 6.1 | 14.3 |
| Smokers who visited their GP | | | | | | | | | | | | |
| Received offer of GP support for smoking cessation | 27.0 | 25.7 | 28.2 | 27.1 | 23.6 | 31.1 | 24.7 | 22.6 | 27.0 | 28.9 | 25.2 | 33.1 |
| High-risk drinkers who visited their GP | | | | | | | | | | | | |
| Received offer of GP support for alcohol reduction | 1.4 | 1.1 | 1.6 | 1.5 | 1.0 | 2.3 | 1.3 | 1.0 | 1.8 | 0.8 | 0.3 | 2.0 |

CI, 95% confidence interval.

[1] Raw weighted estimates aggregated across participants in all survey waves (January 2021 through December 2022).

[2] Data for January 2021, December 2021, and December 2022 are weighted estimates from log-binomial regression with survey month modelled non-linearly using restricted cubic splines (three knots), adjusted for covariates.

40.3% [35.9–45.2] to 49.9% [42.0–59.4], respectively), but no significant change among those who mainly smoked manufactured cigarettes (S6 and S8 Figs).

Other time trends in use of support did not differ significantly by social grade (for either smoking cessation [any paid support $p_{interaction}$ = 0.323; e-cigarettes $p_{interaction}$ = 0.668; evidence-based support $p_{interaction}$ = 0.693] or alcohol reduction [evidence-based support $p_{interaction}$ = 0.482]; S5 and S7 Figs) or main type of cigarettes smoked (smoking cessation only [any paid support $p_{interaction}$ = 0.553; OTC NRT $p_{interaction}$ = 0.083]; S6 Fig).

### Receipt of GP offer of support for smoking cessation and alcohol reduction

Among those who reported visiting their GP in the last 12 months, 27.1% of past-year smokers reported having received an offer of support for smoking cessation and 1.6% of high-risk drinkers reported having received an offer of support for alcohol reduction (Table 2). Between January 2021 and December 2022, there was no significant change in receipt of GP offer of support for smoking cessation or alcohol reduction (Fig 5, Table 2). Time trends in receipt of GP offer of support did not differ significantly by social grade (for smoking cessation [$p_{interaction}$ = 0.120] or alcohol reduction [$p_{interaction}$ = 0.543]; S9 Fig).

### Discussion

A cost-of-living crisis began to affect Great Britain in late 2021 [6]. Between January 2021 (around a year before the start of the crisis) and December 2022 (a year into the crisis), there were few notable changes in the nature of attempts to stop smoking and to reduce alcohol consumption. The proportion of attempts motivated by cost did not change significantly over

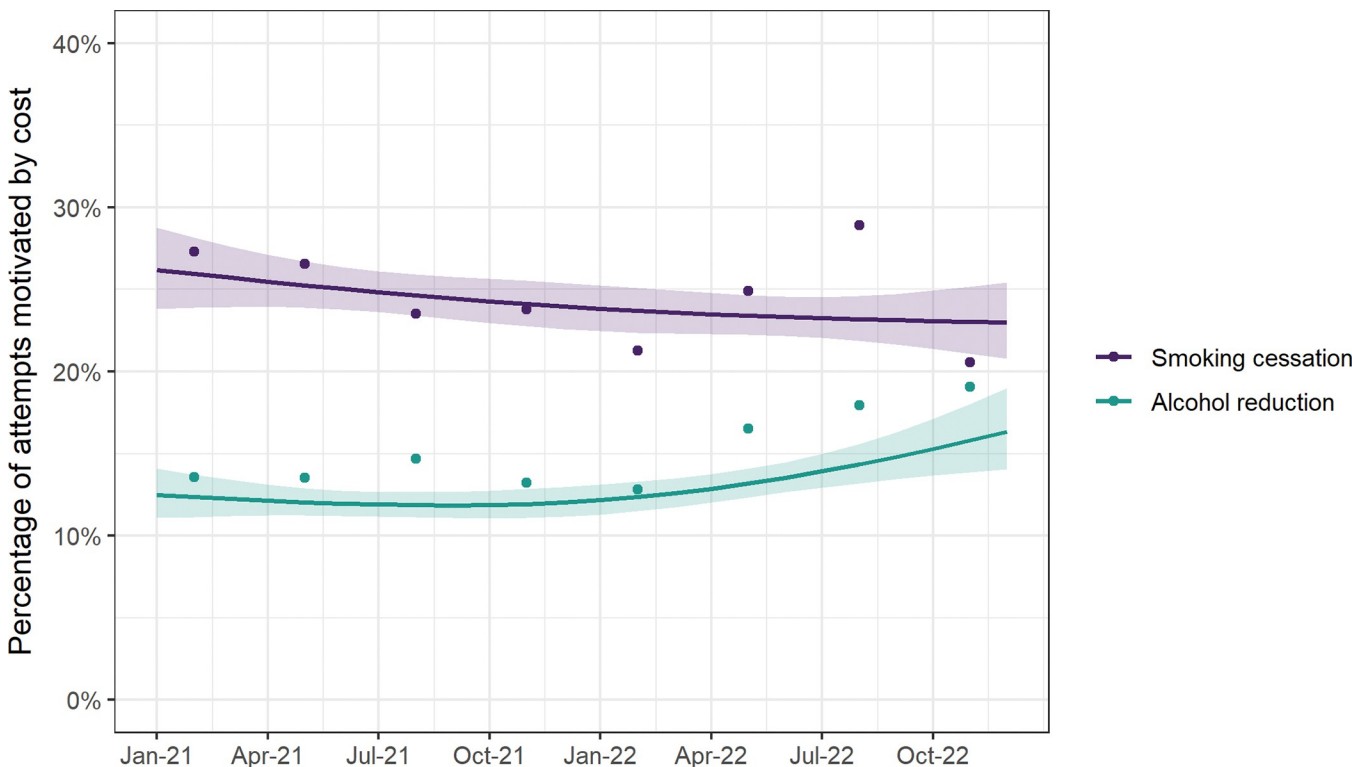

**Fig 1. Time trends in the percentage of attempts to stop smoking and to reduce alcohol consumption that were motivated by cost, January 2021 to December 2022.** Lines represent modelled weighted prevalence by survey month, modelled non-linearly using restricted cubic splines (three knots), adjusting for covariates. Shaded bands represent standard errors. Points represent raw weighted prevalence by quarter.

time among smokers, which may reflect inflation-adjusted expenditure on cigarettes falling between 2020 and 2022 in England [9], but increased during the cost-of-living crisis among high-risk drinkers from less advantaged social grades. The proportion of smokers using any paid support in a quit attempt increased over time, driven by a rise in use of e-cigarettes specifically. Among smokers and high-risk drinkers who visited their GP, there was no significant change in the proportion who received an offer of support.

The proportion of attempts to stop smoking that were motivated by cost was relatively stable over time, with approximately one in four (25%) smokers who made a serious attempt to quit in the past year citing cost as a motive driving their most recent attempt. However, there was some indication in the raw (unmodelled) data that smoking cessation attempts motivated by cost had risen across the first three quarters of 2022, before falling in quarter four. There was a rise in the proportion of attempts to reduce alcohol consumption that were motivated by cost, from around one in eight (12%) to one in six (16%) high-risk drinkers who made an attempt. This was concentrated among high-risk drinkers from less advantaged social grades, among whom the proportion of attempts motivated by cost roughly doubled (from 15% to 30%) since the cost-of-living crisis started in late 2021. We saw no significant increase in overall prevalence of alcohol reduction attempts, indicating that although an increasing proportion of those trying to reduce their consumption say cost is a motivating factor, cost pressures have not yet resulted in a substantial increase in the absolute number of people trying to reduce their alcohol consumption. It will be important to continue to monitor this as the cost-of-living crisis develops. Although the crisis began to affect Great Britain in late 2021 [6], it is plausible that people's motivation to reduce their spending on cigarettes and alcohol has increased

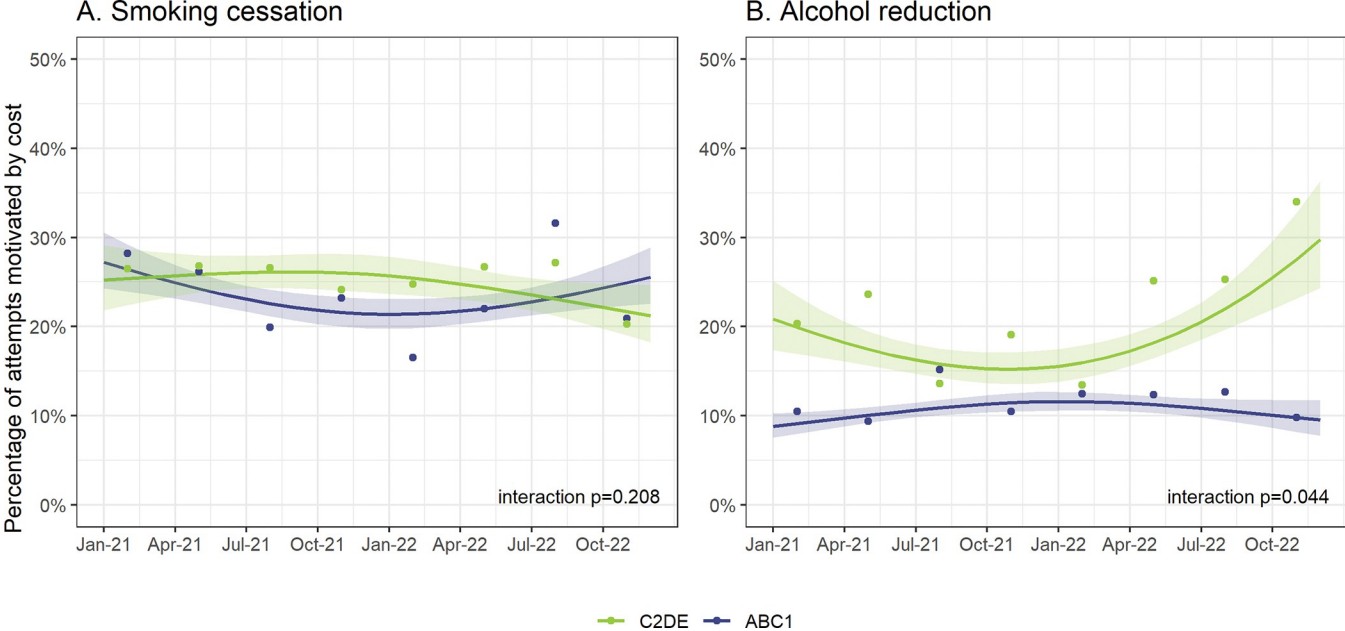

**Fig 2. Time trends in the percentage of attempts to stop smoking and to reduce alcohol consumption that were motivated by cost, January 2021 to December 2022: Interactions with social grade.** Lines represent modelled weighted prevalence by survey month, modelled non-linearly using restricted cubic splines (three knots), adjusting for covariates. Shaded bands represent standard errors. Points represent raw weighted prevalence by quarter. P-values are for the interaction between survey month and social grade.

more in recent months as news coverage of the crisis has increased, inflation and the Bank of England base rate have risen [6, 28], and energy bills have increased substantially over winter [29]. However, a government energy support scheme to help with high energy bills implemented in late 2022 may have reduced some of the financial urgency some people were experiencing for stopping smoking or reducing alcohol consumption. It is also possible that cost motives were starting from a high baseline at the beginning of 2021 caused by financial uncertainty arising from the Covid-19 pandemic (for comparison, 19% of smokers and 9% of high-risk drinkers cited cost as a motive in 2019).

We had hypothesised that financial pressures associated with the cost-of-living crisis may have meant smokers would be less likely to use paid forms of support (i.e., OTC NRT and e-cigarettes). However, we observed an *increase* in use of paid support–specifically e-cigarettes–in a quit attempt increased significantly since the start of the cost-of-living crisis (from 28% in January 2021 to 38% in December 2022). A growing body of evidence shows that e-cigarettes increase the odds a given quit attempt is successful [16, 30], so an increase in their use may translate to improved success rates. Also, while e-cigarettes can be considered a 'paid support' their use over time is still markedly cheaper than cigarette smoking [9], thus increasing the likelihood of switching to these products.

There was no significant change in the use of evidence-based support among smokers trying to quit and high-risk drinkers attempting to reduce their consumption. While it is encouraging not to see any decline, it should be noted that six in ten smokers and nine in ten high-risk drinkers did not use any evidence-based support in their most recent attempt. There is therefore substantial room to improve the success rates of smoking cessation and alcohol reduction attempts by directing people to appropriate support. Interventions that increase uptake of effective support, most of which is available free of charge, could help to ease the financial burden of smoking and alcohol use.

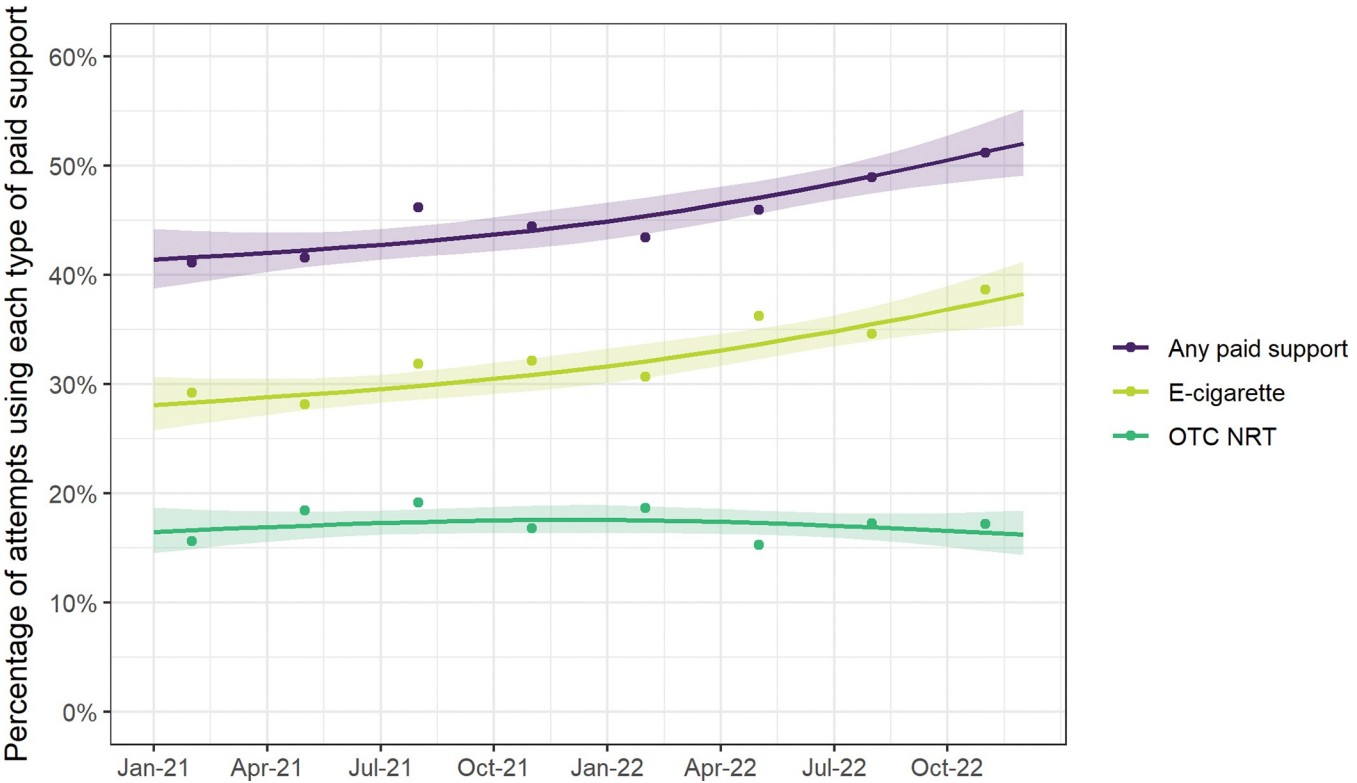

**Fig 3. Time trends in the percentage of attempts to stop smoking that used paid support (overall and by subtype), January 2021 to December 2022.** Lines represent modelled weighted prevalence by survey month, modelled non-linearly using restricted cubic splines (three knots), adjusting for covariates. Shaded bands represent standard errors. Points represent raw weighted prevalence by quarter.

In order to explore whether health professionals have already increased their efforts to help smokers and high-risk drinkers during the cost-of-living crisis, we examined changes in receipt of offers of support for smoking and drinking among those who visited their GP in the last year. Among smokers who visited their GP, 27% reported having received an offer of support for smoking cessation and there was no significant change over time. This is slightly lower than previously observed in England (30.1% [95%CI = 29.1–31.1%] between 2016 and 2019) [31]. Among high-risk drinkers who visited their GP, the proportion who reported receiving an offer of support was much lower, at 1.4%. Previous studies have consistently shown high-risk drinkers to be less likely than smokers to receive a brief intervention, with just 6% reporting receiving any advice or offer of support [32]. That the majority of smokers and drinkers who visited their GP did not report receiving an offer of support suggests opportunities are being missed to boost rates of smoking cessation and alcohol reduction.

Key strengths of the study include the large, representative sample and repeat cross-sectional design. However, there were also limitations. First, the assessment of our outcomes relied upon self-reports, introducing scope for recall bias that may affect estimates of prevalence. However, it is unlikely that recall of motives for quitting, use of support, or receipt of GP offer of support would vary substantially over time, so there should not have been any substantial impact on time trends. Secondly, only people drinking at increasing and higher-risk levels were asked about attempts to reduce alcohol consumption, so our results may not generalise to all drinkers. Thirdly, information on product choice (e.g., brands purchased) was not captured by the survey, so we were unable to determine the extent to which people mitigated

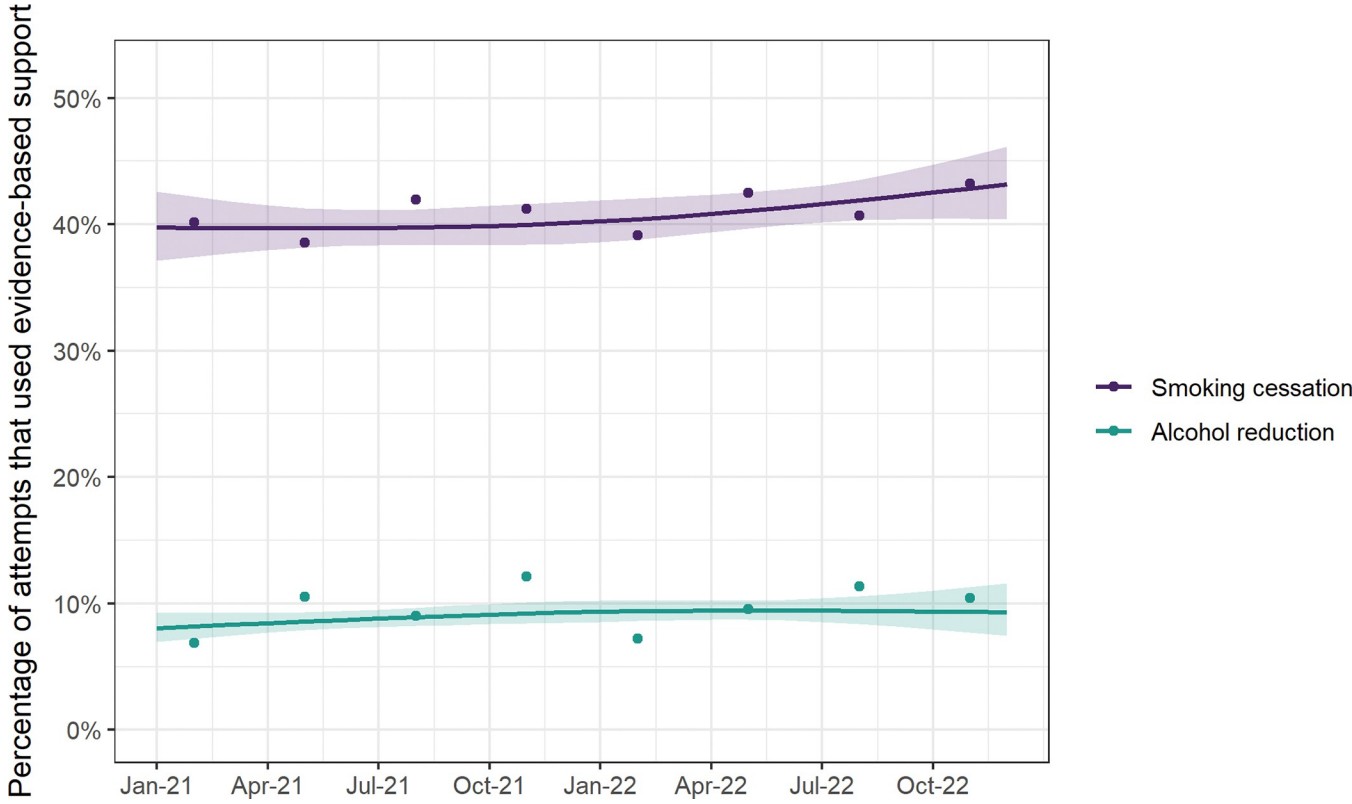

**Fig 4. Time trends in the percentage of attempts to stop smoking and to reduce alcohol consumption that used evidence-based support, January 2021 to December 2022.** Lines represent modelled weighted prevalence by survey month, modelled non-linearly using restricted cubic splines (three knots), adjusting for covariates. Shaded bands represent standard errors. Points represent raw weighted prevalence by quarter.

cost pressures by switching to cheaper products (e.g., from branded alcohol to supermarket own-brands). In addition, we did not analyse changes in cigarette consumption so our data do not offer any insight into whether people are smoking fewer cigarettes in order to reduce their expenditure on tobacco during the cost-of-living crisis. A final limitation is that with only 12 waves of data collected during the cost-of-living crisis to date, this study provides a simple indication of early changes in our outcomes of interest over the first year of the crisis. The optimal design to evaluate the impact of the cost-of-living crisis on these behaviours is an interrupted time-series design, which models the effect of an intervention (in this case, the cost-of-living crisis), taking account of long-term trends in the data. This will not be possible for another year. Considering the financial impact of smoking and drinking [9, 13], particularly on low-income families, in the context of the unstable financial climate, we believed it was important to provide the current initial results, and conduct a more sophisticated time-series analysis when sufficient data points are available to discern any longer-term impact. Further studies could also build upon these results by examining broader impacts of the cost-of-living crisis on smoking and drinking behaviours, such as level of consumption and product choice, to better understand how people are changing their behaviour in response to growing financial pressures. It will also be interesting to monitor the extent to which any changes in behaviour persist when the cost-of-living crisis eases.

In conclusion, there is currently limited evidence of the impact of the cost-of-living crisis in Great Britain on the nature of attempts to stop smoking and to reduce alcohol consumption, or receipt of GP offer of support for smoking cessation and alcohol reduction. There are some

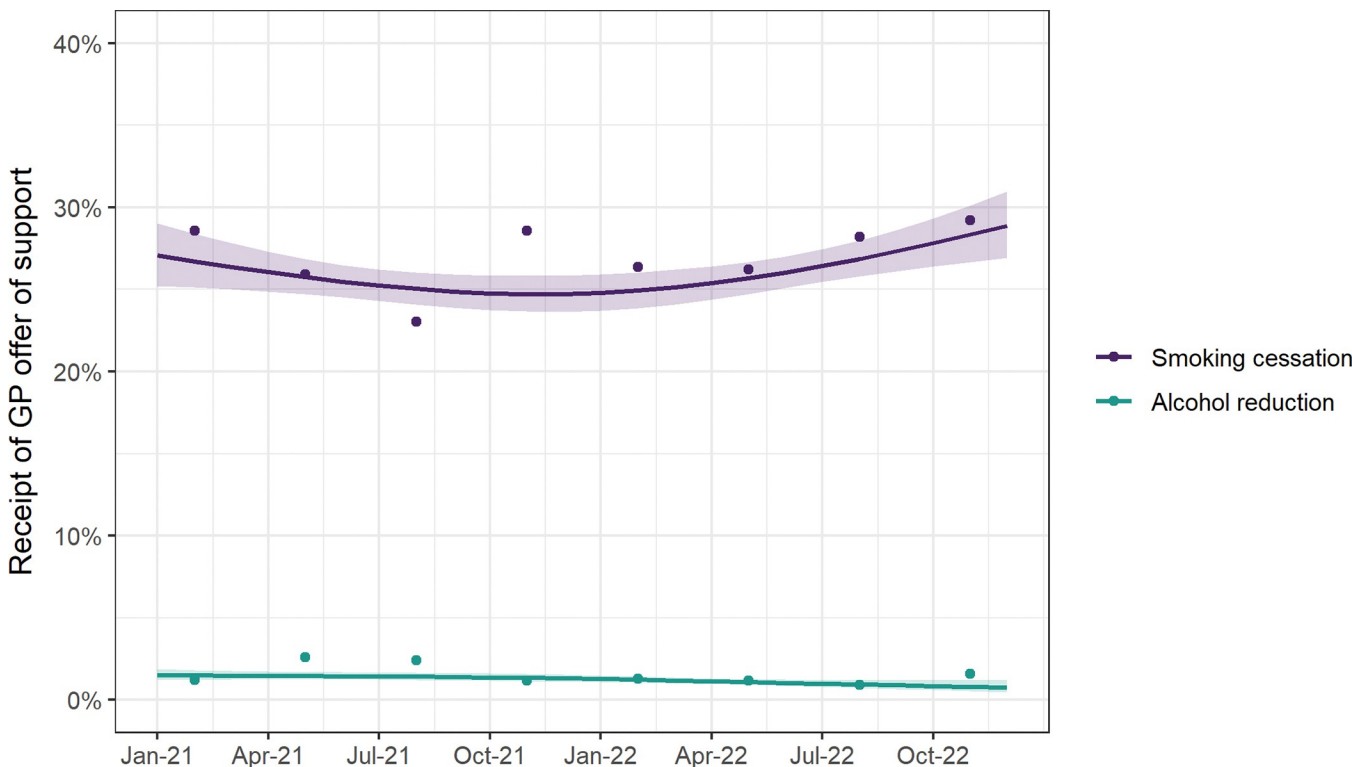

**Fig 5. Time trends in receipt of GP offer of support for smoking cessation and alcohol reduction, January 2021 to December 2022.** Lines represent modelled weighted prevalence by survey month among past-year smokers/high-risk drinkers who visited their GP in the last year, modelled non-linearly using restricted cubic splines (three knots), adjusting for covariates. Shaded bands represent standard errors. Points represent raw weighted prevalence by quarter.

encouraging results: use of evidence-based support has not declined and there has been an increase in the use of e-cigarettes by smokers in quit attempts. However, cost is increasingly motivating alcohol reduction attempts among less advantaged drinkers, and rates of GP offer of support remain low, particularly for high-risk drinkers. It will be important to continue to monitor these trends as the cost-of-living crisis develops in order to identify and respond to any adverse impacts on smokers and high-risk drinkers.

## Supporting information

**S1 Fig. Time trends in the percentage of attempts to stop smoking that were motivated by cost, January 2021 to December 2022: Interaction with main type of cigarettes smoked.** Lines represent modelled weighted prevalence by survey month, modelled non-linearly using restricted cubic splines (three knots), adjusting for covariates. Shaded bands represent standard errors. Points represent raw weighted prevalence by quarter. P-values are for the interaction between survey month and main type of cigarettes smoked.
(TIFF)

**S2 Fig. Time trends in the prevalence of attempts to stop smoking and to reduce alcohol consumption, January 2021 to December 2022.** Lines represent modelled weighted prevalence by survey month, modelled non-linearly using restricted cubic splines (three knots), adjusting for covariates. Shaded bands represent standard errors. Points represent raw weighted prevalence by quarter.
(TIFF)

**S3 Fig. Time trends in the prevalence of attempts to stop smoking and to reduce alcohol consumption, January 2021 to December 2022: Interactions with social grade.** Lines represent modelled weighted prevalence by survey month, modelled non-linearly using restricted cubic splines (three knots), adjusting for covariates. Shaded bands represent standard errors. Points represent raw weighted prevalence by quarter. P-values are for the interaction between survey month and social grade.
(TIFF)

**S4 Fig. Time trends in the prevalence of attempts to stop smoking, January 2021 to December 2022: Interaction with main type of cigarettes smoked.** Lines represent modelled weighted prevalence by survey month, modelled non-linearly using restricted cubic splines (three knots), adjusting for covariates. Shaded bands represent standard errors. Points represent raw weighted prevalence by quarter. P-values are for the interaction between survey month and main type of cigarettes smoked.
(TIFF)

**S5 Fig. Time trends in the percentage of attempts to stop smoking that used paid support (overall and by subtype), January 2021 to December 2022: Interactions with social grade.** Lines represent modelled weighted prevalence by survey month, modelled non-linearly using restricted cubic splines (three knots), adjusting for covariates. Shaded bands represent standard errors. Points represent raw weighted prevalence by quarter. P-values are for the interaction between survey month and social grade.
(TIFF)

**S6 Fig. Time trends in the percentage of attempts to stop smoking that used paid support (overall and by subtype), January 2021 to December 2022: Interactions with main type of cigarettes smoked.** Lines represent modelled weighted prevalence by survey month, modelled non-linearly using restricted cubic splines (three knots), adjusting for covariates. Shaded bands represent standard errors. Points represent raw weighted prevalence by quarter. P-values are for the interaction between survey month and main type of cigarettes smoked.
(TIFF)

**S7 Fig. Time trends in the percentage of attempts to stop smoking and to reduce alcohol consumption that used evidence-based support, January 2021 to December 2022: Interactions with social grade.** Lines represent modelled weighted prevalence by survey month, modelled non-linearly using restricted cubic splines (three knots), adjusting for covariates. Shaded bands represent standard errors. Points represent raw weighted prevalence by quarter. P-values are for the interaction between survey month and social grade.
(TIFF)

**S8 Fig. Time trends in the percentage of attempts to stop smoking that used evidence-based support, January 2021 to December 2022: Interaction with main type of cigarettes smoked.** Lines represent modelled weighted prevalence by survey month, modelled non-linearly using restricted cubic splines (three knots), adjusting for covariates. Shaded bands represent standard errors. Points represent raw weighted prevalence by quarter. P-values are for the interaction between survey month and main type of cigarettes smoked.
(TIFF)

**S9 Fig. Time trends in receipt of GP offer of support for smoking cessation and alcohol reduction, January 2021 to December 2022: Interactions with social grade.** Lines represent modelled weighted prevalence by survey month, modelled non-linearly using restricted cubic splines (three knots), adjusting for covariates. Shaded bands represent standard errors. Points

represent raw weighted prevalence by quarter. P-values are for the interaction between survey month and social grade.
(TIFF)

## Author Contributions

**Conceptualization:** Sarah E. Jackson, Sharon Cox, Lion Shahab, Jamie Brown.

**Data curation:** Jamie Brown.

**Formal analysis:** Sarah E. Jackson.

**Funding acquisition:** Lion Shahab, Jamie Brown.

**Investigation:** Sarah E. Jackson, Sharon Cox, Lion Shahab, Jamie Brown.

**Methodology:** Sarah E. Jackson, Sharon Cox, Lion Shahab, Jamie Brown.

**Resources:** Jamie Brown.

**Supervision:** Jamie Brown.

**Visualization:** Sarah E. Jackson.

**Writing – original draft:** Sarah E. Jackson.

**Writing – review & editing:** Sharon Cox, Lion Shahab, Jamie Brown.

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
