## [Decision Letter · Decision Letter 0]

4 Apr 2023

PONE-D-23-05249Impact of the cost-of-living crisis on the nature of attempts to stop smoking and to reduce alcohol consumption in Great Britain: a representative population survey, 2021-2022PLOS ONE

Dear Dr. Jackson,

Thank you for submitting your manuscript to PLOS ONE. After careful consideration, we feel that it has merit but does not fully meet PLOS ONE’s publication criteria as it currently stands. Therefore, we invite you to submit a revised version of the manuscript that addresses the points raised during the review process.

We look forward to receiving your revised manuscript.

Kind regards,

Anil Gumber, Ph.D.

Academic Editor

PLOS ONE

Journal Requirements:

"I have read the journal's policy and the authors of this manuscript have the following competing interests: JB has received unrestricted research funding from Pfizer and J&J, who manufacture smoking cessation medications. LS has received honoraria for talks, an unrestricted research grant and travel expenses to attend meetings and workshops from Pfizer, and has acted as paid reviewer for grant awarding bodies and as a paid consultant for health care companies. All authors declare no financial links with tobacco companies, e-cigarette manufacturers, the alcohol industry, or their representatives."

5. Please amend your list of authors on the manuscript to ensure that each author is linked to an affiliation. Authors’ affiliations should reflect the institution where the work was done (if authors moved subsequently, you can also list the new affiliation stating “current affiliation:….” as necessary).

Additional Editor Comments:

Thanks for investigating an important topic on impact of cost-of-living on smoking and alcohol behaviour. There is a need to undertake a sub-group analysis for heavy smokers (e.g. smoking 20 or more cigarettes). Authors have defined excessive drinking category as a sub-group but not considered for smokers group. I was surprised that number of daily cigarettes smoked data has not been shown. impact of Cost-of-living must have impacted on the reduction of number of cigarettes. There is a problem in identifying reduction in rolled cigarettes due to rising prices/inflation. I don't know whether the authors have looked into a greater details the reduction in expenses on cigarettes especially the rolling ones or they might have shifted to e-cigarettes due to cost. One of the reviewers has suggested what about including rising prices or inflation in the analysis or building a scenario to current level of inflation which has shooted to >10% in the discussion also.

Reviewers' comments:

Reviewer's Responses to Questions

**Comments to the Author**

1. Is the manuscript technically sound, and do the data support the conclusions?

Reviewer #1: Yes

Reviewer #2: Yes

2. Has the statistical analysis been performed appropriately and rigorously? 

Reviewer #1: Yes

Reviewer #2: Yes

3. Have the authors made all data underlying the findings in their manuscript fully available?

Reviewer #1: No

Reviewer #2: Yes

4. Is the manuscript presented in an intelligible fashion and written in standard English?

Reviewer #1: Yes

Reviewer #2: Yes

5. Review Comments to the Author

Reviewer #1: This paper explores a very important question on the impact of the cost-of-living crisis on attempts to quit smoking and reduce alcohol consumption. The methodology is exceedingly clear, and the broader literature is well referenced. The aims are clearly stated, and the methods are well constructed to address the aims. The conclusions of the paper are well supported by the results. The paper concludes that there have been no major impacts of the cost-of-living crisis on attempts to reduce smoking and alcohol consumption. I only suggest a major revision based on a single comment which has the potential to change the conclusions of the paper. The comment is as follows:

The trend for the weighted raw prevalence of smokers attempting to quit (Figure 1) appears to show an increase until the last quarter data point. This trend appears for both the ABC1 and C2DE groups (Figure 2. Panel A). There is also a milder version of this trend for C2DE high-risk drinkers (Figure 2. Panel B). The energy support scheme started in late 2022 which may have provided a reprieve for households and postponed any attempt to quit smoking/drinking. Price inflation of cigarettes and alcohol may be an important factor in the decision to quit smoking/reduce alcohol consumption, but this price inflation has been steadily rising for many years. Energy prices rose sharply in the first three quarters of 2022 (even before winter, as the price per unit rose) and represent a significant proportion of household budgets. This may have pushed people to reduce their spending in other areas, such as cigarettes and alcohol. It may be worth exploring this by evaluating in which month does the prevalence of quitting attempts starts to decline from the previous quarter. If the month-by-month analysis shows that quitting attempts closely follow energy prices, this may offer some evidence that rise in costs – not of cigarettes or alcohol – may have an impact on quitting attempts.

I also have some minor comments for the authors to address:

Introduction/Background

- Pg 3: The average weekly expenditure on cigarettes is based on the population of smokers, whereas the average weekly expenditure on alcohol is based on the whole of UK households. ONS estimated in 2017 that 20% of British adults do not drink, so the average weekly expenditure may be underestimated here.

- Pg 4 – 2nd paragraph: “In particular, any impact is likely to have been greater among disadvantaged groups”. Could you add “socioeconomically disadvantaged”? You use this phrase later on, so I think it’s important to use from the beginning.

- Pg 4 – 2nd paragraph: “While overall alcohol consumption (…) and experience more harm (3,18)”. Could you write “experience more health harm” or “experience higher mortality risk”.

- Pg 4 – 3rd paragraph: You mention that “It is therefore plausible that any impact of the cost-of-living crisis on quit attempts may be more pronounced among smokers who mainly smoke manufactured cigarettes” because manufactured cigarettes are more expensive than hand-rolled. However, studies have shown that people already “downgrade” from manufactured to hand-rolled as the price of the former rises (https://bmjopen.bmj.com/content/5/6/e007697.short). It may also be plausible that people already smoking hand-rolled cigarettes are more vulnerable to cost pressures, and therefore more likely to quit smoking as the cost of living increases.

Results

- Pg 10 “High-risk drinkers surveyed in England in May, July, September, and November 2022 (n=2,076) were ineligible for inclusion in our analysis as they were not asked questions on our outcomes of interest.” Could you elaborate a little on this? Why is the data missing? Is there no data at all on any outcome of interest for these months to at least have a form of comparison with the main analysis and its validity?

- Pg 12 – Would it be possible to include time trends for people quitting/reducing alcohol without a cost motive? This would be helpful to frame the results.

- Table 1 - Why is there no data for “mainly smokes hand-rolled cigarettes for smokers who visited GP?

Discussion

- Pg 20 – Paragraph 3. Please re-phrase “A growing body of evidence shows that e-cigarettes increase the chances a given quit attempt is successful” to something like “A growing body of evidence shows that e-cigarettes increase the odds that a quitting attempt is successful”.

- Pg 22 – Paragraph 1. If there is evidence of the energy prices being an important factor, then an ITS design with more data may present more problems, given the changes in policy (energy support scheme entering in last quarter in 2022 and it’s ending sometime this year). Therefore, your design with early data may be more informative.

Overall

PLOS One requires that if “there are restrictions on publicly sharing data—e.g. participant privacy or use of data from a third party—those must be specified.” – Apologies if I missed it, but there was no statement specifying these restrictions within the manuscript.

Reviewer #2: This manuscript by Jackson et al is a manuscript is an original and interesting piece of research that highlights the cost of living on the nature and rate of smoking and alcohol cessation attempts. It is clear that the data has been collected with high academic standard and rigour. There are a few comments attached that Jackson at al may want to consider but I also do not object to the manuscript being published in the current state that it is in.

I look forward to seeing this manuscript in print and more data that is collected in the future.

6. PLOS authors have the option to publish the peer review history of their article (what does this mean?). If published, this will include your full peer review and any attached files.

Reviewer #1: **Yes: **Ana Correa

Reviewer #2: No

---

## [Author Response · Author response to Decision Letter 0]

2 May 2023

Additional editor comments

Thanks for investigating an important topic on impact of cost-of-living on smoking and alcohol behaviour. There is a need to undertake a sub-group analysis for heavy smokers (e.g. smoking 20 or more cigarettes). Authors have defined excessive drinking category as a sub-group but not considered for smokers group. I was surprised that number of daily cigarettes smoked data has not been shown. impact of Cost-of-living must have impacted on the reduction of number of cigarettes. There is a problem in identifying reduction in rolled cigarettes due to rising prices/inflation. I don't know whether the authors have looked into a greater details the reduction in expenses on cigarettes especially the rolling ones or they might have shifted to e-cigarettes due to cost. One of the reviewers has suggested what about including rising prices or inflation in the analysis or building a scenario to current level of inflation which has shooted to >10% in the discussion also.

Response: The reason we have analysed data from high-risk drinkers as opposed to all adult drinkers is to focus on at-risk populations in terms of health outcomes: any level of smoking and high-risk levels of alcohol consumption (but not lower levels) are known to pose significant risk to health. These groups are therefore high priority from a public health perspective. In addition, the questions assessing our outcomes of interest (e.g., motives for alcohol reduction attempts) are only asked of participants who report drinking at higher-risk levels (AUDIT-C ≥5). Thus, this is not a sub-group analysis, but rather the most complete analysis we can undertake for drinkers. We now make this clearer in the manuscript:

“We were only able to analyse drinking outcomes among high-risk drinkers (as opposed to all adult drinkers) because our outcomes of interest were not assessed in participants reporting low or moderate consumption (i.e., AUDIT-C score <5).”

We did not model trends in cigarettes per day because from a public health perspective, reduction in alcohol consumption is a desirable outcome and so is quitting smoking (but reducing cigarette consumption is not). The focus of this pre-registered paper was on the nature of attempts to stop smoking/reduce alcohol consumption specifically, rather than smoking and alcohol behaviour more broadly. We appreciate that there may be interest in how the cost-of-living crisis has affected other smoking and drinking behaviours not assessed here – of which there are many – and will consider undertaking a detailed analysis of these in a separate paper. We now include some additional text on directions for future studies in the discussion:

“Further studies could also build upon these results by examining broader impacts of the cost-of-living crisis on smoking and drinking behaviours, such as level of consumption and product choice, to better understand how people are changing their behaviour in response to growing financial pressures. It will also be interesting to monitor the extent to which any changes in behaviour persist when the cost-of-living crisis eases.”

Reviewer 1

This paper explores a very important question on the impact of the cost-of-living crisis on attempts to quit smoking and reduce alcohol consumption. The methodology is exceedingly clear, and the broader literature is well referenced. The aims are clearly stated, and the methods are well constructed to address the aims. The conclusions of the paper are well supported by the results. The paper concludes that there have been no major impacts of the cost-of-living crisis on attempts to reduce smoking and alcohol consumption. I only suggest a major revision based on a single comment which has the potential to change the conclusions of the paper. The comment is as follows:

The trend for the weighted raw prevalence of smokers attempting to quit (Figure 1) appears to show an increase until the last quarter data point. This trend appears for both the ABC1 and C2DE groups (Figure 2. Panel A). There is also a milder version of this trend for C2DE high-risk drinkers (Figure 2. Panel B). The energy support scheme started in late 2022 which may have provided a reprieve for households and postponed any attempt to quit smoking/drinking. Price inflation of cigarettes and alcohol may be an important factor in the decision to quit smoking/reduce alcohol consumption, but this price inflation has been steadily rising for many years. Energy prices rose sharply in the first three quarters of 2022 (even before winter, as the price per unit rose) and represent a significant proportion of household budgets. This may have pushed people to reduce their spending in other areas, such as cigarettes and alcohol. It may be worth exploring this by evaluating in which month does the prevalence of quitting attempts starts to decline from the previous quarter. If the month-by-month analysis shows that quitting attempts closely follow energy prices, this may offer some evidence that rise in costs – not of cigarettes or alcohol – may have an impact on quitting attempts.

Response: Given the relatively small sample sizes surveyed in each month, the monthly data are quite noisy – so there are fluctuations up and down across the entire study period. It is therefore difficult to pinpoint the exact month in which the proportion citing cost as a motive begins to decline. For example, for smoking cessation attempts, the lower CI for the raw weighted Q3 in 2022 was 23.8% and upper CI for the raw weighted Q4 was 25.1%. It is for this reason that we present quarterly (rather than monthly) estimates as data points on our figures. However, we have added to the results text to highlight the potential levelling off of an increasing trend in the fourth quarter of 2022 to bring this pattern to the reader’s attention:

On smoking cessation attempts: “However, the raw weighted data suggested a potential increasing trend between the first and third quarters of 2022, before falling in quarter four (Figure 1).”

…

On alcohol reduction attempts: “The raw weighted data showed a similar pattern to smoking cessation attempts, with some indication of levelling off in the fourth quarter of 2022 (Figure 1) – at least among those from social grades ABC1 (Figure 2).”

We also raise it in the discussion:

“…there was some indication in the raw (unmodelled) data that smoking cessation attempts motivated by cost had risen across the first three quarters of 2022, before falling in quarter four.”

…

“However, a government energy support scheme to help with high energy bills implemented in late 2022 may have reduced some of the financial urgency some people were experiencing for stopping smoking or reducing alcohol consumption.”

I also have some minor comments for the authors to address:

Introduction/Background

- Pg 3: The average weekly expenditure on cigarettes is based on the population of smokers, whereas the average weekly expenditure on alcohol is based on the whole of UK households. ONS estimated in 2017 that 20% of British adults do not drink, so the average weekly expenditure may be underestimated here.

Response: We have edited this sentence to read:

“Data from the 2020/21 Living Costs and Food Survey show the average UK household spends £11 a week on alcohol (13) (which likely underestimates drinkers’ expenditure, given ~20% of UK adults do not drink at all (14)), indicating similar potential to make savings by reducing alcohol consumption.”

- Pg 4 – 2nd paragraph: “In particular, any impact is likely to have been greater among disadvantaged groups”. Could you add “socioeconomically disadvantaged”? You use this phrase later on, so I think it’s important to use from the beginning.

Response: We have edited as suggested.

- Pg 4 – 2nd paragraph: “While overall alcohol consumption (…) and experience more harm (3,18)”. Could you write “experience more health harm” or “experience higher mortality risk”.

Response: We have edited to ‘experience higher mortality risk’, in line with the reference cited.

- Pg 4 – 3rd paragraph: You mention that “It is therefore plausible that any impact of the cost-of-living crisis on quit attempts may be more pronounced among smokers who mainly smoke manufactured cigarettes” because manufactured cigarettes are more expensive than hand-rolled. However, studies have shown that people already “downgrade” from manufactured to hand-rolled as the price of the former rises (https://bmjopen.bmj.com/content/5/6/e007697.short). It may also be plausible that people already smoking hand-rolled cigarettes are more vulnerable to cost pressures, and therefore more likely to quit smoking as the cost of living increases.

Response: Good point – we have added the following:

“However, it may also be the case that people already smoking hand-rolled cigarettes are more vulnerable to cost pressures, and therefore may be more likely to try to quit smoking as the cost of living increases.”

Results

- Pg 10 “High-risk drinkers surveyed in England in May, July, September, and November 2022 (n=2,076) were ineligible for inclusion in our analysis as they were not asked questions on our outcomes of interest.” Could you elaborate a little on this? Why is the data missing? Is there no data at all on any outcome of interest for these months to at least have a form of comparison with the main analysis and its validity?

Response: We have clarified the reason for this:

“High-risk drinkers surveyed in England in May, July, September, and November 2022 (n=2,076) were ineligible for inclusion in our analysis as they were not asked questions on our outcomes of interest (alcohol questions were only assessed in alternate waves over this period due to availability of funding).”

- Pg 12 – Would it be possible to include time trends for people quitting/reducing alcohol without a cost motive? This would be helpful to frame the results.

Response: We have run the models for all smoking cessation and alcohol reduction attempts and display the results in Supplementary Figures 2-4. We refer to these in the results section to provide a comparison with the cost motive results:

“This rise was specific to alcohol reduction attempts that were motivated by cost, with no such increase observed when we analysed trends in all alcohol reduction attempts over this period (Supplementary Figures 2-4).”

And in the discussion:

“We saw no significant increase in overall prevalence of alcohol reduction attempts, indicating that although an increasing proportion of those trying to reduce their consumption say cost is a motivating factor, cost pressures have not yet resulted in a substantial increase in the absolute number of people trying to reduce their alcohol consumption.”

- Table 1 - Why is there no data for “mainly smokes hand-rolled cigarettes for smokers who visited GP?

Response: We have added this figure to the table.

Discussion

- Pg 20 – Paragraph 3. Please re-phrase “A growing body of evidence shows that e-cigarettes increase the chances a given quit attempt is successful” to something like “A growing body of evidence shows that e-cigarettes increase the odds that a quitting attempt is successful”.

Response: We have changed ‘chances’ to ‘odds’.

- Pg 22 – Paragraph 1. If there is evidence of the energy prices being an important factor, then an ITS design with more data may present more problems, given the changes in policy (energy support scheme entering in last quarter in 2022 and it’s ending sometime this year). Therefore, your design with early data may be more informative.

Response: We have edited this section to clarify that an ITS design would offer insight into any longer-term impacts on behaviour:

“Considering the financial impact of smoking and drinking (9,13), particularly on low-income families, in the context of the unstable financial climate, we believed it was important to provide the current initial results, and conduct a more sophisticated time-series analysis when sufficient data points are available to discern any longer-term impact.”

Overall

PLOS One requires that if “there are restrictions on publicly sharing data—e.g. participant privacy or use of data from a third party—those must be specified.” – Apologies if I missed it, but there was no statement specifying these restrictions within the manuscript.

Response: We have now uploaded all data used in these analyses to Open Science Framework and provided a link in the manuscript.

Reviewer 2

This manuscript by Jackson et al is a manuscript is an original and interesting piece of research that highlights the cost of living on the nature and rate of smoking and alcohol cessation attempts. It is clear that the data has been collected with high academic standard and rigour. There are a few comments attached that Jackson at al may want to consider but I also do not object to the manuscript being published in the current state that it is in.

I look forward to seeing this manuscript in print and more data that is collected in the future.

Introduction, paragraph 1:

“Although concerns for health are usually the primary motive for changing these behaviours, cost is also an important factor for many smokers and high-risk drinkers”

Cost may be a factor, depending largely on what type of alcohol people drink. Bargain shops such as Home Bargains and B+M (mainly located in the North of England) make alcohol relatively cheap and as a result, cost may become less of a factor. 

Response: We now refer to there being differences in cost between products in paragraph 2:

“However, it is also possible that people have switched to cheaper products to reduce their expenditure without changing their consumption.”

Introduction, paragraph 2: 

“The cost-of-living crisis refers to a fall in ‘real’ disposable incomes (that is, adjusted for inflation and after taxes and benefits) that people living in Great Britain have experienced since late 2021” 

One thing to consider is that the cost of living crisis has been going on for many years prior to 2021. Only now is the cost of living crisis appearing to impact the upper- and middle-class families and households. Many working class families have been having issues with household budgets from around 2010. 

Response: Thank you for this important comment. We have edited to:

“The cost-of-living crisis refers to a fall in ‘real’ disposable incomes (that is, adjusted for inflation and after taxes and benefits) that people living in Great Britain have experienced since late 2021 (6) – although many working class families have struggled with household budgets for much longer than this (7).”

Introduction, paragraph 2: 

“The two most popular aids used by smokers trying to quit are e-cigarettes…”

Do disposable vape pods also come under this or are they their own category entirely? Often times these are cheaper than the traditional e-cigarettes.

Response: This includes all e-cigarettes, including disposable vapes.

Covariates:

Employment status would be a useful covariate too. COVID meant that many lost their jobs and current cost of living means that individuals may also lose their jobs/businesses.

Response: We did not include employment status as a covariate on the basis that it may mediate changes in our outcomes over time, and thereby adjustment may have obscured the associations we planned to assess.

---

## [Editor Report · Decision Letter 1]

9 May 2023

PONE-D-23-05249R1Impact of the cost-of-living crisis on the nature of attempts to stop smoking and to reduce alcohol consumption in Great Britain: a representative population survey, 2021-2022PLOS ONE

Dear Dr. Jackson,

Thank you for submitting your manuscript to PLOS ONE. After careful consideration, we feel that it has merit but does not fully meet PLOS ONE’s publication criteria as it currently stands. Therefore, we invite you to submit a revised version of the manuscript that addresses the points raised during the review process.

We look forward to receiving your revised manuscript.

Kind regards,

Anil Gumber, Ph.D.

Academic Editor

PLOS ONE

Journal Requirements:

Additional Editor Comments:

Thanks for incorporating partly suggestions by the reviewers. I think the expenditure on smoking and its reduction due to Cost of living crisis is an important issue which you have not covered. Also we know price-elasticity or income-elasticity could be a determinantal components under cost-of-living crisis. Basic economic theory and consumer behaviour also highlights principles the substitution effects, i.e. if income reduces then consumer tends to reduce consumption of the particular good or move to a cheaper options (e.g. brom branded alcohol to local supermarket branded ones, same is the case for smoking). Your paper failed to address this behaviour. My suggestions are to provide a detailed limitations of your study section by answering what factors are supported from the data and what aspects are not covered due to lack of information in the dataset.

Reviewers' comments:

While revising your submission, please upload your figure files to the Preflight Analysis and Conversion Engine (PACE) digital diagnostic tool, https://pacev2.apexcovantage.com/. PACE helps ensure that figures meet PLOS requirements. To use PACE, you must first register as a user. Registration is free. Then, login and navigate to the UPLOAD tab, where you will find detailed instructions on how to use the tool. If you encounter any issues or have any questions when using PACE, please email PLOS at figures@plos.org. Please note that Supporting Information files do not need this step.<quillbot-extension-portal></quillbot-extension-portal>

---

## [Author Response · Author response to Decision Letter 1]

10 May 2023

Journal Requirements:

Response: We have checked our reference list and confirm it is complete and correct.

Additional Editor Comments:

Thanks for incorporating partly suggestions by the reviewers. I think the expenditure on smoking and its reduction due to Cost of living crisis is an important issue which you have not covered. Also we know price-elasticity or income-elasticity could be a determinantal components under cost-of-living crisis. Basic economic theory and consumer behaviour also highlights principles the substitution effects, i.e. if income reduces then consumer tends to reduce consumption of the particular good or move to a cheaper options (e.g. brom branded alcohol to local supermarket branded ones, same is the case for smoking). Your paper failed to address this behaviour. My suggestions are to provide a detailed limitations of your study section by answering what factors are supported from the data and what aspects are not covered due to lack of information in the dataset.

Response: Thank you for raising these issues. We have expanded our limitations section to address them:

“Thirdly, information on product choice (e.g., brands purchased) was not captured by the survey, so we were unable to determine the extent to which people mitigated cost pressures by switching to cheaper products (e.g., from branded alcohol to supermarket own-brands). In addition, we did not analyse changes in cigarette consumption so our data do not offer any insight into whether people are smoking fewer cigarettes in order to reduce their expenditure on tobacco during the cost-of-living crisis.”s

Reviewers' comments:

[None]

---

## [Editor Report · Decision Letter 2]

11 May 2023

Impact of the cost-of-living crisis on the nature of attempts to stop smoking and to reduce alcohol consumption in Great Britain: a representative population survey, 2021-2022

PONE-D-23-05249R2

Dear Dr. Jackson,

We’re pleased to inform you that your manuscript has been judged scientifically suitable for publication and will be formally accepted for publication once it meets all outstanding technical requirements.

Kind regards,

Anil Gumber, Ph.D.

Academic Editor

PLOS ONE

Additional Editor Comments (optional):

Thanks for incorporating limitations of the study in terms of dataset gap on curtailment of expenditure on alcohol and smoking as well as price substitution effects due to cost-of-living crisis.

Reviewers' comments:

<quillbot-extension-portal></quillbot-extension-portal>

---

## [Editor Report · Acceptance letter]

15 May 2023

PONE-D-23-05249R2 

Impact of the cost-of-living crisis on the nature of attempts to stop smoking and to reduce alcohol consumption in Great Britain: a representative population survey, 2021-2022 

Dear Dr. Jackson:

I'm pleased to inform you that your manuscript has been deemed suitable for publication in PLOS ONE. Congratulations! Your manuscript is now with our production department. 

Kind regards, 

on behalf of

Dr. Anil Gumber 

Academic Editor

PLOS ONE